# Integrity of the DNA and Cellular Ultrastructure of Cryptoendolithic Fungi in Space or Mars Conditions: A 1.5-Year Study at the International Space Station

**DOI:** 10.3390/life8020023

**Published:** 2018-06-19

**Authors:** Silvano Onofri, Laura Selbmann, Claudia Pacelli, Jean Pierre de Vera, Gerda Horneck, John E. Hallsworth, Laura Zucconi

**Affiliations:** 1Department of Ecological and Biological Sciences, University of Tuscia, 01100 Viterbo, Italy; onofri@unitus.it (S.O.); pacelli@unitus.it (C.P.); zucconi@unitus.it (L.Z.); 2Italian National Antarctic Museum (MNA), Mycological Section, 16166 Genoa, Italy; 3German Aerospace Center (DLR) Berlin, Institute of Planetary Research, Rutherfordstreet 2, 12489 Berlin, Germany; jean-pierre.devera@dlr.de; 4German Aerospace Centre, Institute of Aerospace Medicine, Linder Hoehe, D 51170 Köln, Germany; Gerda.Horneck@dlr.de; 5Institute for Global Food Security, School of Biological Sciences, MBC, Queen’s University Belfast, Belfast BT9 7BL, UK; j.hallsworth@qub.ac.uk

**Keywords:** cryptoendolithic black fungi, DNA and cellular damage, EXPOSE-E, LIFE experiment, space exposure and Mars conditions, ionizing- and ultra-violet radiation

## Abstract

The black fungi *Cryomyces antarcticus* and *Cryomyces minteri* are highly melanized and are resilient to cold, ultra-violet, ionizing radiation and other extreme conditions. These microorganisms were isolated from cryptoendolithic microbial communities in the McMurdo Dry Valleys (Antarctica) and studied in Low Earth Orbit (LEO), using the EXPOSE-E facility on the International Space Station (ISS). Previously, it was demonstrated that *C. antarcticus* and *C. minteri* survive the hostile conditions of space (vacuum, temperature fluctuations, and the full spectrum of extraterrestrial solar electromagnetic radiation), as well as Mars conditions that were simulated in space for a 1.5-year period. Here, we qualitatively and quantitatively characterize damage to DNA and cellular ultrastructure in desiccated cells of these two species, within the frame of the same experiment. The DNA and cells of *C. antarcticus* exhibited a higher resistance than those of *C. minteri*. This is presumably attributable to the thicker (melanized) cell wall of the former. Generally, DNA was readily detected (by PCR) regardless of exposure conditions or fungal species, but the *C. minteri* DNA had been more-extensively mutated. We discuss the implications for using DNA, when properly shielded, as a biosignature of recently extinct or extant life.

## 1. Introduction

Microorganisms able to tolerate or thrive in habitats which are lethal or detrimental for most terrestrial life-forms are defined as extremophiles. Adaptations of microbial systems to severe conditions in natural Earth environments are of astrobiological significance. They can act as models for searching for life beyond Earth, or to investigate the possibility of life transfer between planets [1,2]. In view of space exploration missions focused on identifying evidence for extant or extinct life in the Solar System and beyond, several experiments, ground-based on Earth or in space, have been performed in the last decades [3,4,5,6]. Simulating space conditions in ground experiments is very useful for testing the resistance of extremophilic microorganisms and biomolecules to specific space parameters, i.e., radiation and vacuum, and combinations of extremes. The complexity of the space environment (microgravity, vacuum, and full solar radiation), however, cannot be reliably reproduced in the laboratory [7]. Therefore, space missions are necessary for testing the ability of organisms and their macromolecular systems to survive and resist the stresses that would be experienced during interplanetary transfer. Microorganisms and biomolecules have been exposed in space, e.g., in the BIOPAN exposure facility on the FOTON capsule, where lichens survived a two-week exposure to the Low Earth Orbit (LEO) environment [8]; lichens and cyanobacteria survived a 10-day exposure to space during the Lithopanspermia experiment [9,10]; and lichens, Antarctic fungi, bacteria, and cyanobacteria survived 16–18 months in space on the European Technology Exposure Facility’s (EuTEF) “EXPOSE-E” and “EXPOSE-R” experimental systems on the International Space Station [11,12,13].

The two black fungi *Cryomyces antarcticus* and *Cryomyces minteri* are cryptoendolithic and micro-colonial species that occur within sandstone outcrops of the McMurdo Dry Valleys of Antarctica (Southern Victoria Land) [14]. This environment is typically dry, experiences sub-zero temperatures, and is exposed to ultra-violet radiation, so has been used as a terrestrial analogue for martian environments. These two species of *Cryomyces* are endemic to the Antarctic, and both are psychrophilic, grow very slowly, and belong to the class Dothideomycetes, but do not yet have phylogenetic designations at the levels of Order or Family [14]. They both have an idiosyncratic morphology, which includes melanized cell walls, much more evident in *C. antarcticus*, with visible incrustations (Figure 1). Especially when dehydrated, they have an outstanding ability to resist a wide number of stresses including ultra-violet and ionizing radiation [2], extremes of temperatures, and desiccation-rehydration events [15,16,17,18,19,20,21,22,23]. *Cryomyces* species are used as a eukaryotic models for testing the plausibility of life transfer inside meteorites (Lithopanspermia theory), life detection on Mars, and other astrobiological questions. They have been used in various space experiments, demonstrating their extraordinary ability to resist both ground-based [17,20,21] and space experiments [18,19,24].

During a long-term exposure experiment performed outside of the Columbus module of the International Space Station (ISS), known as the Lichens and Fungi Experiment (LIFE), *C. antarcticus* and *C. minteri* were placed on the European Space Agency’s (ESA’s) EXPOSE-E facility and exposed to six different conditions. Some fungal samples were fully exposed to space, other samples were partially or completely shielded from solar radiation, and others were exposed to Mars conditions simulated in space (Table 1).

A set of control samples was kept on Earth, so we refer to these as the Ground controls. During the mission, the sun-exposed LIFE samples received up to 1879 eSCh (estimated Solar Constant hours) (1572 eSCh in Mars conditions) [12]. *C. antarcticus*, retrieved after a 1.5-year exposure, showed a reasonably good survival rate; with 12.5% of cells viable according to cultivation tests (CFUs), and up to 80% of cells with intact membranes (by PMA assay) when exposed to space conditions. Samples exposed to Mars conditions simulated in space indicated that 8.4% of cells were viable (from CFUs) and 65.0% had intact membranes. *C. minteri* was much more affected by the exposure, both in terms of membrane damage and survival [18,19].

These results were of special interest in terms of Lithopanspermia, since they concern rock-dwelling organisms; rocks may, in fact, supply an additional external protection to face the impact-driven ejection into space [27,28], the transfer from one planet to another, and the capture by—and landing onto—another planet [27,29,30]. A terrestrial rock-habitat could also enable lithobionts to be transported between planets in streams of space dust, as recent research suggests [31].

Here, we report an investigation of the resilience of DNA and cellular ultrastructure of *C. antarcticus* and *C. minteri* after a 1.5-year exposure to space and simulated Mars conditions, during the LIFE study. The specific aims were to: (i) investigate the resistance of, or damage to, their genomic DNA; (ii) characterize any changes in cellular ultrastructure that are indicative of stress, damage, or lethality; and (iii) determine whether DNA is sufficiently stable to act as a biosignature for life detection [20,21,32].

## 2. Materials and Methods 

### 2.1. Spaceflight Data of LIFE

The EXPOSE-E is part of the ESA’s European Technology Exposure Facility (EuTEF, ESA) (Figure 2A), which was designed for testing different materials under selected parameters of space. The European Technology Exposure Facility (including EXPOSE-E which contained the LIFE experiment) was launched to the ISS on the Space Shuttle STS-122 on 7 February 2008 (Figure 2B). EXPOSE-E was then mounted on the outer side of the ISS’s Columbus module on 15 February 2008 (Figure 2C), decommissioned on 1 September 2009, and retrieved on 2 September 2009 via extravehicular activity. EXPOSE-E was returned to Earth on the Space Shuttle STS-128 on 12 September 2009. During the 1.5-year exposure time, the fungi were subjected to the treatments detailed in Table 1. Ground controls were mantained at room temperature (25 °C) in the laboratory of Systematic Botany and Mycology (University of Tuscia, Viterbo, Italy) for the same time-period (1.5 years).

### 2.2. Biological Test Systems of LIFE 

Biomass of *C. antarcticus* CCFEE 515 and *C. minteri* CCFEE 5187 was obtained by first cultivating on malt extract agar for three months at 15 °C, until the colonies, near-spherical, attained about a 1 mm-diameter. Cells where dehydrated gradually, over a period of 20 days at 25 °C in silica-gel desiccators [17,18], to stabilize the samples before flight. This also mitigates against potential damage that can be caused by the more-rapid dehydration that can be induced by a space vacuum. An additional set of samples was prepared following the same procedure described above, preserved in dried conditions in a desiccator at room temperature (25 °C) in the laboratory of Systematic Botany and Mycology (University of Tuscia, Viterbo, Italy), and used as the Ground controls.

### 2.3. DNA Extraction and PCR 

DNA was extracted from four dried colonies, using the NucleoSpin^®^ Plant II kit (Macherey-Nagel, Düren, Germany) following the protocol optimized for fungi. Internal Transcribed Spacer (ITS) and Large SubUnit (LSU) amplifications were performed using BioMix™ (BioLine GmbH, Luckenwalde, Germany), by adding 1 μL (5 pM) of each primer to 0.1 ng of template DNA, making it up to a final volume of 25 μL using distilled water. The amplification was carried out using MyCycler™ Thermal Cycler (Bio-Rad Laboratories GmbH, Munich, Germany) equipped with a heated lid.

The rRNA genes were amplified as follows: for the ITS region, the first denaturation step (2 min at 95 °C) was followed by denaturation for 30 s at 95 °C, annealing for 30 s at 55 °C, and extension for 30 s at 72 °C. For the LSU region, the first denaturation step (3 min at 95 °C) was followed by denaturation for 45 s at 95 °C, annealing for 30 s at 52 °C, and extension for 3 min at 72 °C. These last three steps were carried out 35 more times, with an additional final extension for 5 min at 72 °C for ITS and 7 min at 72 °C for LSU. Primers ITS5 and ITS4 [33], and LR5 and LR7 [34] were used to amplify ITS and LSU rDNA portions, respectively. Band intensity was determined and compared using ImageJ software Version 2 (https://imagej.nih.gov/ij/download.html) [35].

### 2.4. Sequencing and Alignment

Sequence analyses were carried out to evaluate the mutational load of the gene target. Sequencing reactions were carried out by the Macrogen Online Sequencing Order System (Macrogen, Seoul, Korea), Pathfinder in Genomics Research, 1001 World Meridian Center, Seoul, Korea (http://www.macrogen.com); electropherograms visualization and proof-reading were performed using Chromas 2.6.4 (Technelysium Pty Ltd., Brisbane, Australia). The mutational load of the DNA was evaluated as the measure of damage to the macromolecule, by aligning the sequences using Multiple Sequence Comparison by Log-Expectation (MUSCLE). In this analysis, DNA from Fresh colonies was used as the reference (not mutated sequences), in addition to Ground controls. The amplitude of the DNA damage, in terms of mutational load, was evaluated by aligning the sequences using MUSCLE software (www.ebi.ac.uk/tools/msa/muscle/) in Molecular Evolutionary Genetics Analysis Version 6.0 (MEGA6) [36]. Based on the alignments, dendrograms were constructed using the Unweighted Pair Group Method with the Arithmetic Mean (UPGMA) option in MEGA6. These data were then plotted as a graph, based on the assumption that the distance among the analyzed sequences is proportional to the DNA mutations.

### 2.5. Random Amplification of Polymorphic DNA Assay

Random Amplification of Polymorphic DNA (RAPD) was used to analyze the DNA damage sustained by the whole genome. It was performed using BioMix™ (BioLine GmbH, Luckenwalde, Germany), by adding 1 μL (5 pM) primer to 1 ng template DNA and making it up to a final volume of 25 μL using distilled water. The primer used for RAPD was GGAGGAGGAGGAGGAGGAGGA (GGA_7_) [37]. Amplification was carried out using an initial denaturation step for 2 min at 94 °C, followed by denaturation for 20 s at 94 °C, annealing for 60 s at 49 °C, and extension for 20 s at 72 °C. The last three steps were each repeated 40 times, with a last extension for 6 min at 72 °C. Molecular approaches based on PCR-stop (see above) and genomic PCR fingerprinting assays are powerful tools for evaluating DNA damage; the protocols used here were validated for microorganisms [3,20,38,39] and optimized for fungi [22]. These approaches have been already applied to assess DNA damage in a number of studies carried out on both ground-based and space experiments on dried colonies of *Cryomyces* [20,21,24].

### 2.6. Quantitative PCR

QuantitativePCR (q-PCR) allows quantification of the intact DNA copies and it has been applied on *Cryomyces* in this study for the first time. Following DNA extraction and purification, q-PCR (Bio-Rad CFX96 Touch™ Real Time PCR Detection System) was used to determine the number of fungal ITS ribosomal DNA fragments. Five μL of purified genomic DNA (0.2 ng/µL) was added to 12 μL of PCR cocktail containing 1× Power Sybr-Green PCR Master Mix (Applied Bios, Foster City, CA, USA), as well as NS91 forward primer (5′-GTC CCT GCC CTT TGT ACA CAC-3′) [40] and ITS4 reverse primer (5′-TCCTCCGCTTATTGAATATGC-3′), each at a final concentration of 5 pM. Sterile distilled water was added to reach a final volume of 25 μL. These primers amplified a portion of the 201 bp as part of 500 bp which spans the entire ITS region of rRNA encoding genes. Reactions were run in triplicate with the following program: 2 min at 95 °C, followed by 35 cycles 30 s at 95 °C (denaturing), 30 s at 55 °C (annealing), and 30 s at 72 °C (elongation). A melting-curve analysis followed, with temperature increments of 0.5 °C (from 60 to 90 °C). All tests were performed in triplicate, and means and standard deviations were calculated and plotted. Statistical analyses were performed by one-way analysis of variance (ANOVA) and the pair wise multiple comparison procedure (*t* test), carried out using the statistical software SigmaStat Version 2.0 (Jandel Scientific Software, San Jose, CA, USA).

### 2.7. Transmission Electron Microscopy

The ultrastructural integrity was investigated by Transmission Electron Microscopy (TEM). Ground controls and exposed samples were treated with glutaraldehyde 5% *v*/*v* in cacodylate sucrose buffer 0.1 M (0.1 M saccarose; 5 mM CaCl_2_, 5 mM MgCl_2_, pH 7.2) for 12 h at 4 °C, washed three times in the same buffer for 1 h each at 4 °C, and fixed with 1% OsO_4_
*w*/*v* + 0.15% Ruthenium red *w*/*v* in 0.1 M cacodylate buffer (pH 7.2) for 3 h at 4 °C. Samples were washed in distilled water (two times, for 30 min at 4 °C), treated with 1% *w*/*v* uranyl acetate in distilled water for 1 h at 4 °C, and washed using water (two times, for 30 min each time at 4 °C). Samples were dehydrated in ethanol solutions: 30%, 50%, 70% *v*/*v* (15 min each, at 25 °C), and pure ethanol (1 h at 25 °C), and permeated in mixtures of pure ethanol. LR White resin (Agar Scientific, Essex, UK) (2:1 for 3 h; 1:1 for 3 h, 1:2 overnight), in a rotator, at 4 °C, and in pure resin for 36 h, was applied as a final step until complete penetration of the mixture. Samples were then embedded in pure resin at 48–52 °C for 48 h and blocks were cut by Ultramicrotome (Reichert-Jung E Ultracut, Wien, Austria) using a diamond knife. Ultra-thin sections (60–80 nm) were placed onto copper grids and stained with uranyl acetate 2% *w*/*v* and lead citrate 0.2% *w*/*v*. They were examined using a JEOL 1200 EX II Transmission Electron Microscope. The images were captured using a CCD Camera SIS VELETA (Olympus, Münster, Germany) using the iTEM software Version 2009 (EMSIS GmbH, Münster, Germany).

## 3. Results

### 3.1. DNA Integrity, Assessed via Single-Gene Amplification

The quality of amplification (ITS and LSU) was determined by electrophoresis on an agarose gel (Figure 3) because the production of long DNA segments increases the chances of detecting genomic damage. DNA damage was evaluated based on the assumption that the band intensity was proportional to the extent of mutations on the template [22].

A 700 bp fragment, corresponding to the ITS portion, was amplified from *C. antarcticus* after the exposure to space and Mars conditions, and there was no marked difference in band intensity relative to the ground controls (Figure 3A). By contrast, amplification of two longer portions of the same genes (1600–2000 bp) revealed clear differences. For those from fungal samples exposed to space conditions (solar ultra-violet 100% irradiance or in the dark) and simulated Mars conditions in the dark, intensity was reduced (Figure 3B,C). More specifically, for the 2000 bp ITS-LSU fragment, there was a 20% decrease in band intensity for both Space 100% irradiation and Mars dark samples, compared with the space-exposed dark treatments. Band intensity was reduced in the controls kept on Earth for the duration of the experiment. Yet, despite the decreases in band intensity in some of the samples exposed to the most-threatening conditions, DNA amplification was always successful.

For *C. minteri*, damage was apparent even for the shortest gene sequence tested (ITS, 700 bp), where both Mars-conditions dark-treatment and the 100% irradiated space samples yielded bands of lower intensities; with a 20% decrease for the 100% irradiated space treatment when compared with the space-exposed dark treatment (Figure 3D). The greatest differences in band intensity were apparent for the longest fragments ITS-LSU (2000 bp) (Figure 3F). There was no band corresponding to 100% irradiated space conditions, whereas bands were present for dark and 100% irradiated (>200 nm UV) treatment in the simulated martian atmosphere, and their intensities were very low.

### 3.2. DNA Integrity Assessment by Sequencing, Aligning and UPGMA Analysis

Amplicon sequencing of all the samples under different treatments of *C. antarcticus* gave reproducible electropherograms (Figure 4).

For *C. minteri*, these gene portions were not readable due to overlapping peaks (Figure 5). The most-extensive changes in nucleotide sequences (implying the greatest degree of DNA damage) were observed in samples exposed to 100% irradiation either for space exposure or under Mars conditions.

The differences in the mutational load between the two species and among different treatments were calculated on the base of the alignments (not shown) and highlighted in the UPGMA dendrograms (Figure 6A,B). In *C. antarcticus*, the mutational load in the DNA did not exceed 5% compared to Fresh colonies, regardless of the treatment. In the UPGMA analysis, rooted with the same gene sequence from the *C. minteri* Ground control to highlight the group, sequences from both the Fresh colonies and Ground controls, and all the sequences from different treatments, pooled together (Figure 6A).

By contrast, the mutational load of *C. minteri* was high and, according to the electropherograms, in accordance with the intensity of the treatments. The percentage of mutations in Ground control, Space dark, Space 0.1% irradiation, and Mars 0.1% irradiation was round 0.6% compared to Fresh colonies, while it increased up to 15% for Mars dark and up to 73% for Space 100% irradiation and Mars 100% irradiation. The high number of mutations, deletions, and insertions in the sequences was evident from the sequence alignments (not shown), with an increasing degree of mutation found in the samples exposed to the most-threatening conditions. In the UPGMA dendrogram (Figure 6B), the sequences from Ground control, Fresh colony, Space dark, and Space 0.1% irradiation treatments grouped together, while the others were progressively more distant according to increases in irradiation intensity. The treatment that had sustained the highest DNA damage was the Mars 100% irradiation sample, which was used as the outgroup.

### 3.3. Whole Genome Integrity: Insights from Molecular Fingerprinting 

For both *Cryomyces* species, molecular fingerprint profiles obtained by RAPD are shown in Figure 7. It is evident that *C. antarcticus* profiles were well reproducible in all conditions (Figure 7A). For *C. minteri*, a very faint profile of bands was obtained for Space 100% irradiation, when compared with those for other treatments (Figure 7B).

### 3.4. DNA Damage According to Quantitative PCR

For *C. antarcticus*, there were no significant differences in the number of copies of the ITS target gene in any treatment relative to the Ground control, except for a slight increase in Space 100% irradiation (Figure 8A). For *C. minteri*, the number of DNA copies yielded was significantly lower in the Space 100% irradiation, simulated Mars dark conditions, and Mars 100% irradiation (>200 nm UV) than in the ground control (Figure 8B).

### 3.5. Cellular Ultrastructure as Determined by Transmission Electron Microscopy

Transmission electron micrographs of the *C. antarcticus*—Ground controls (Figure 9A,B) revealed intact cell membranes, but some plasmolysis was apparent (Figure 9A, arrow); lipid bodies were present (Figure 9B, arrow). Some cells of the Space 100% irradiation samples exhibited plasmolysis (Figure 9C,D, arrows). However, damage had been sustained by some cells as evidenced by scattered fragments of cell walls (Figure 9E, arrows) and an amorphous cytoplasm (Figure 9F, arrow).

By contrast, there were signs of extensive damage in the *C. minteri* cells exposed to the Space 100% irradiation. Micrographs showed some undamaged cells in these samples (Figure 9I), but the majority exhibited amorphous cytoplasm (Figure 9J, arrow) and broken cell walls (Figure 9K,L, arrows). Cells of the *C. minteri* Ground control (Figure 9G,H) were generally characterized by well-preserved cell membranes (Figure 9H); yet a small number of cells with discontinuous cell-membranes and cell-wall fragments were also visible (Figure 9G, arrow). 

## 4. Discussion

The possible existence of life on other planets and the capacity to survive space irradiation, potentially allowing some microorganisms to pass unimpeded through an interplanetary transfer and disseminate throughout the universe, is one of the main questions in Astrobiology. Another main concern is to define significantly distinguishable biosignatures, indicating the presence of putative extinct or even extant life beyond Earth [41,42], an issue representing a central focus and a challenge for the upcoming rover missions on Mars.

From this perspective, resistance of the Antarctic fungi *Cryomyces antarcticus* CCFEE 515 and *C. minteri* CCFEE 5187 were tested in ground-based experiments, and in space (in the LIFE experiment), for the first time [18,19]. Both earlier Earth laboratory and space experiments revealed a superior endurance of *C. antarcticus*. Namely, it was almost three times more resistant than *C. minteri* to UV-B radiation at the maximum dose applied [22]. Its higher survival capacity (12.5%) compared to *C. minteri* (0.5%) was also assessed after 1.5 years in space in the LIFE experiment [18].

Here, we reported findings arising from PCR-based assays and TEM approaches to detect DNA and ultrastructural damage in *C. antarcticus* and *C. minteri* from the LIFE experiment. Results showed that solar radiation in space was the most threatening factor in terms of cellular and macromolecular (DNA) damage for the two polyextremophilic fungi studied. This was most evident for *C. minteri*, where (i) band intensities decreased in proportion to the dose applied in the single gene amplification assay until complete disappearance of the longer amplicons (Figure 3E,F); (ii) mutations covered 73%, even in the shortest gene length (ITS); (iii) the q-PCR assay gave a decrement of the DNA copies yielded of one order magnitude in Space 100% irradiation, compared to the Ground control (Figure 8B); and (iv) there was a clear bleaching of the profiles at Space 100% irradiation (not observed in any other treatment), as revealed by the RAPD assay. 

By contrast, this phenomenon was less evident in *C. antarcticus*; there was no significant difference relative to the ground control for some treatments. We observed that: (i) in the single-gene amplification assay, it was always possible to get products, even for the longest gene; (ii) the mutational load never exceeded 5%, even at the highest treatment; (iii) according to q-PCR assays, any decrease of the DNA copy-number yielded in all samples (Figure 8A); and (iv) RAPD profiles were perfectly maintained in all treatments. Although solar radiation was the most-biologically hostile factor, simulated Mars conditions also inflicted considerable damage to samples. In some cases, there was a clear reduction in DNA amplification in *C. minteri*, as ascertained by single gene amplification (Figure 3E,F) and q-PCR (Figure 8B). We hypothesize that it is the oxidizing conditions of the martian atmosphere that inflict damage on microbial DNA and, presumably, on other cellular macromolecules [43]. We cannot currently explain the higher band intensity observed for *C. minteri* Mars 100% irradiation treatment compared to Mars dark treatment; besides, the mutational load accumulated was much higher in the first compared to the latter (73% and 15%, respectively).

Evidence from the current study proves that *C. antarcticus* is considerably more resilient to space and Mars conditions than *C. minteri*, and this is consistent with the higher survival rates of the former to these treatments reported in earlier studies (see above). *C. antarcticus* and *C. minteri* are phylogenetically closely related and morphologically similar, with heavily melanized cell walls and a similar ecophysiology; both are cryptoendoliths characterized by slow growth rates, with optimal growth at 15 °C [14]. The mechanistic basis of the high levels of resilience observed in *C. antarcticus* may, however, relate to its thicker cell wall: more melanin is present due to the greater quantity and depth of the cell-wall material (Figure 1 and Figure 9). The protective effect of melanin against ultra-violet radiation and ionizing radiation is well established, for all types of biological systems [44,45]. This pigment can absorb radiation energy, dissipate it as heat while limiting the generation of reactive oxygen species (ROS), and/or also absorbs ROS, thus preventing or minimizing potential damage to cellular macromolecules, trapping and neutralizing the free radicals or ROS generated by the ionization of molecules [46,47,48,49,50]. It is likely, therefore, that the thicker melanin layer of *C. antarcticus* is the primary characteristic of its cellular phenotype and stress biology that may have enabled a superior performance during the 1.5-year exposure to extreme treatments in the current study.

Recent studies compared the effects of both densely and sparsely ionizing radiations on melanized *C. antarcticus* and artificially not-melanized *C. antarcticus* (treated with tricyclazole) cells [51,52], revealing that melanin has a potent protective role in this species. Considering the permanent dehydration of space condition in the present experiment, DNA repair systems remain inactive, so melanin may act as the primary defense against DNA damage [22,48]. 

Data in the current study indicated an unexpectedly high level of damage in Ground controls of both species, sometimes even higher than in treated samples. Besides, this situation is rather frequent and has been encountered in other space (or simulated space) experiments carried out using a range of microbial systems (Billi; Rettberg, pers. comm.). One inescapable hypothesis arising from this finding is that the de-hydrated state in space can act as a force for the preservation of microbial cells. We also speculate that, in the partially dehydrated ground-control samples, a minimal level of chemical/metabolic activity takes place and, in the presence of atmospheric oxygen plus traces of water, oxygen reactive molecules (ROS) may be produced. At this low metabolic rate, cells are presumably inefficient in their response to oxidative stress [38].

Space conditions/events cannot be faithfully reproduced in the laboratory, and cellular stress or damage experienced under such conditions cannot be meaningfully extrapolated [7]. Microbial experiments in space can, therefore, provide invaluable insights into the ability of microbial cells to survive interplanetary transits. Besides, the effect of a single component of the complex cosmic radiation cannot be extrapolated. For this reason, the irradiation campaign of the Starlife consortium was planned to simulate and study the effect of different types of ionizing radiations (γ-rays and heavy ions), representing a major part of the galactic cosmic radiation spectrum [53], and further investigations have been carried out on Earth. In the Starlife project, a high resistance in terms of the survival and molecular and ultrastructural integrity of *C. antarcticus* was demonstrated for both ionizing radiations (up to 56 kGy) and accelerated He ions (up to 1 kGy) [2,21].

The phenomenon that a terrestrial microbe (*C. antarcticus*) retains undamaged DNA after prolonged exposure to space and Mars conditions is consistent with our hypothesis that space can preserve biological material. It also indicates that DNA can have value as a biosignature for life detection [21,39], assuming that a putative extraterrestrial life form resembles terrestrial cellular systems and has comparable adaptations and phenotypic traits to those of *C. antarcticus*. Although it is still uncertain whether DNA can remain intact over thousands to millions of years, on or near the surface of Mars, there may be some geochemical conditions that can enhance its longevity (e.g., some brines). Further work is needed to see how biotic and geochemical factors can interact to preserve DNA or other biosignatures for life. Given that cellular stress during physiological conditions, and cellular damage during microbial dormancy, can be qualitatively different phenomena [54], more work is also needed to establish the biophysical limits for the biotic activity of *C. antarcticus* and *C. minteri*. Automated in situ life-detection instruments, such as the Search for Extra-Terrestrial Genomes (SETG) instrument, are now under development [43], integrating nucleic acid extraction and nanopore sequencing. As extracellular DNA may be bound to minerals or destroyed once hydrated, and to optimize nucleic acid detection, a new irradiation campaign will be carried out by the Starlife consortium. Moreover, melanin as a protective screen against space radiation may have value for protecting bioregenerative systems and astronauts in future space-exploration missions [2,48].

## Figures and Tables

**Figure 1 life-08-00023-f001:**
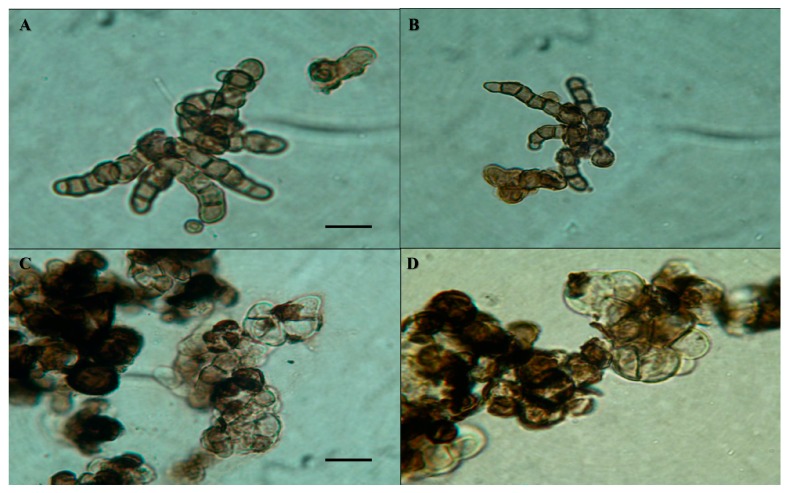
Cells of *Cryomyces antarcticus* (**A**,**B**) and *Cryomyces minteri* (**C**,**D**) after three months incubation on malt extract agar at 15 °C. Panels (**A**,**C**) were reproduced with permission from Selbmann et al. [14]. Photographs were taken using light microscopy, and scales bars = 10 μm.

**Figure 2 life-08-00023-f002:**
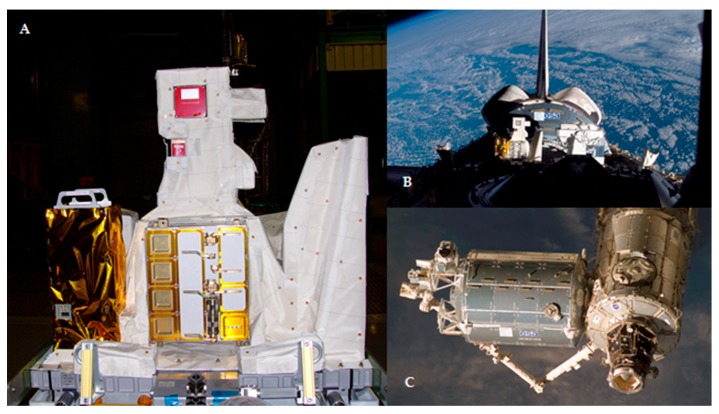
(**A**) Expose-E integrated in the EuTEF platform, at the Kennedy Space Center, before flight; (**B**) EXPOSE-E fitted onto the Space Shuttle; and (**C**) robotic arm in action during assembly of the EuTEF on the ISS.

**Figure 3 life-08-00023-f003:**
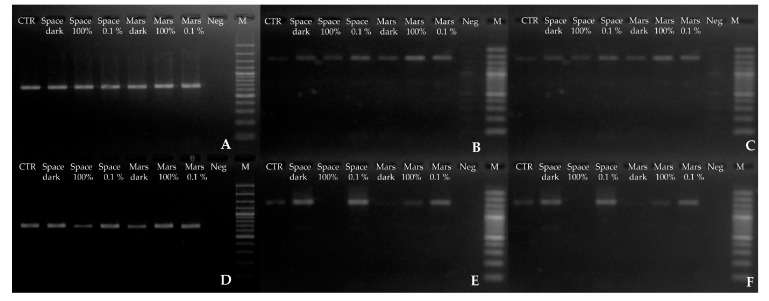
DNA bands obtained by single gene PCR. *C. antarcticus* (**A**) 700 bp; (**B**) 1600 bp; (**C**) 2000 bp and *C. minteri*; (**D**) 700 bp; (**E**) 1600 bp; (**F**) 2000 bp. CTR = Ground control; Space dark = space-exposed dark treatment; Space 100% = space-exposed 100% irradiation; Space 0.1% = space-exposed 0.1% irradiation; Mars dark = simulated Mars conditions, dark treatment; Mars 100% = simulated Mars conditions 100% irradiation; Mars 0.1% = simulated Mars conditions 0.1% irradiation (see Table 1); Neg = negative control and M = DNA ladder.

**Figure 4 life-08-00023-f004:**
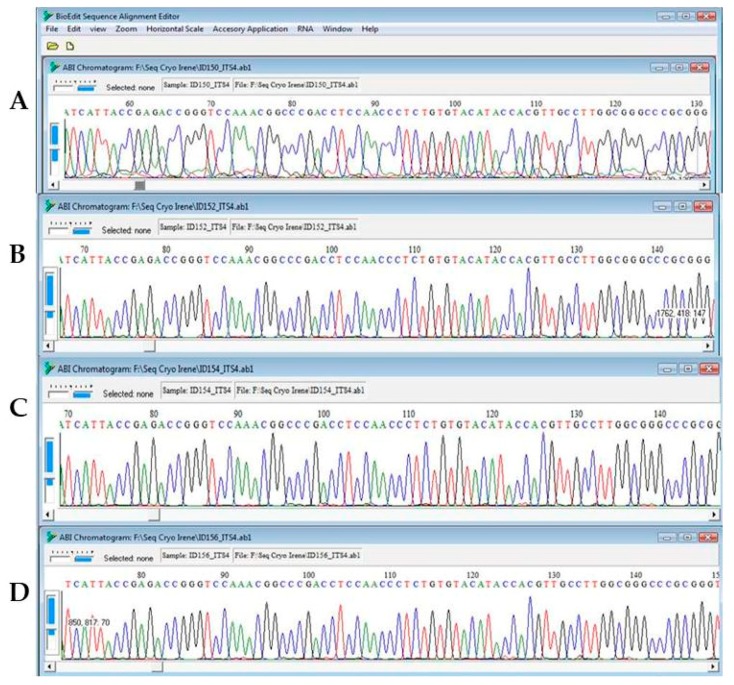
Electropherograms showing nucleotide sequences for the ITS (60–135 position) of *C. antarcticus*: (**A**) Ground control; (**B**) Space 100% irradiation; (**C**) simulated Mars dark conditions; and (**D**) simulated Mars 100% irradiation.

**Figure 5 life-08-00023-f005:**
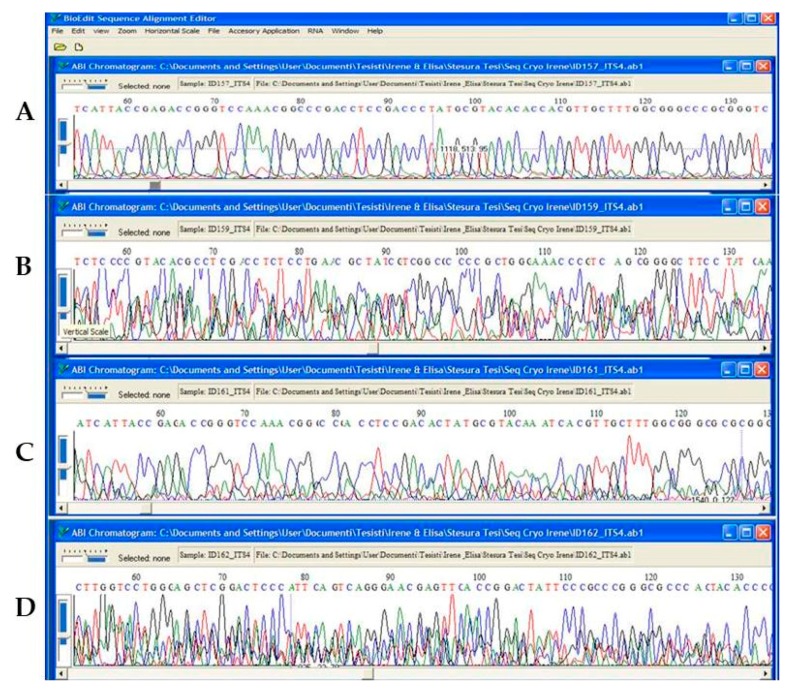
Electropherograms showing nucleotide sequences for the ITS (60–135 position) of *C. minteri*. (**A**) Ground control; (**B**) Space 100% irradiation; (**C**) dark Mars; and (**D**) Mars 100% irradiation.

**Figure 6 life-08-00023-f006:**
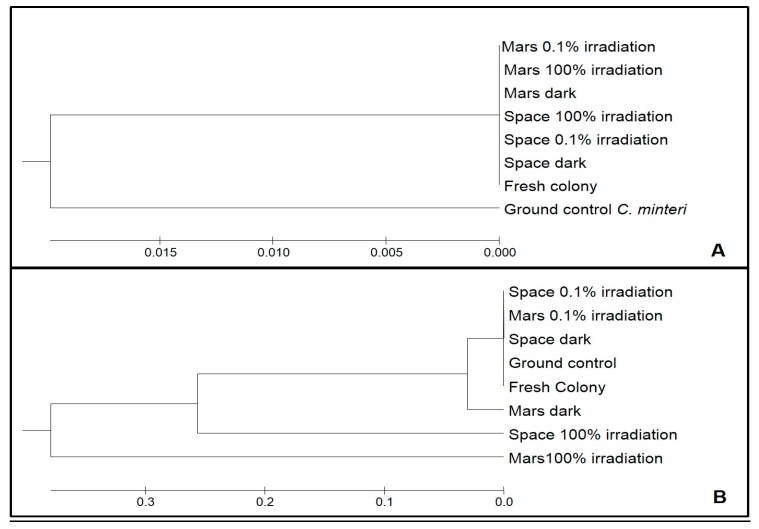
Unweighted Pair Group Method with Arithmetic Mean (UPGMA) analyses based on *C. antarcticus* sequences alignment; the dendrogram was rooted with the same gene sequence from the Ground control of *C. minteri* (**A**); UPGMA analyses based on *C. minteri* sequences alignment; the dendrogram rooted with Mars 100% irradiation resulted in the most mutated sequence after treatment (**B**).

**Figure 7 life-08-00023-f007:**
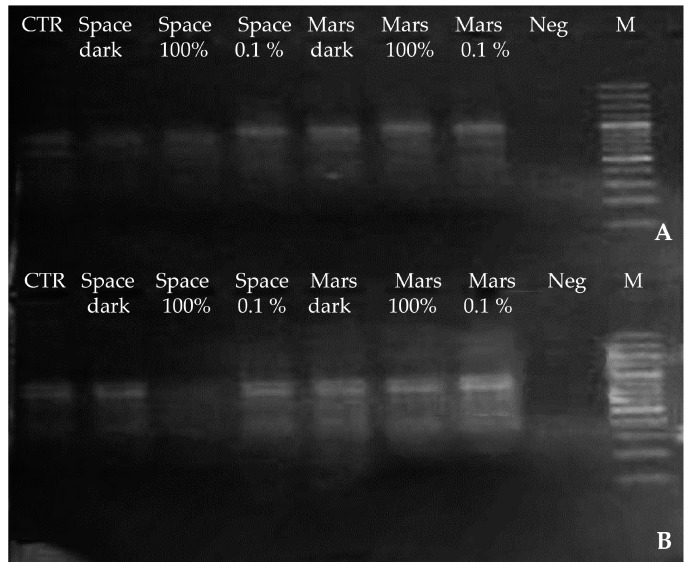
Genomic DNA fingerprint profiles from the RAPD assay of experimental samples (see Table 1) for: (**A**) *C. antarcticus* and (**B**) *C. minteri*. CTR = Ground control; Space dark = space-exposed dark treatment; Space 100% = space-exposed 100% irradiation; Space 0.1% = space-exposed 0.1% irradiation; Mars dark = simulated Mars conditions, dark treatment; Mars 100% = simulated Mars conditions 100% irradiation; Mars 0.1% = simulated Mars conditions 0.1% irradiation; Neg = negative control and M = DNA ladder.

**Figure 8 life-08-00023-f008:**
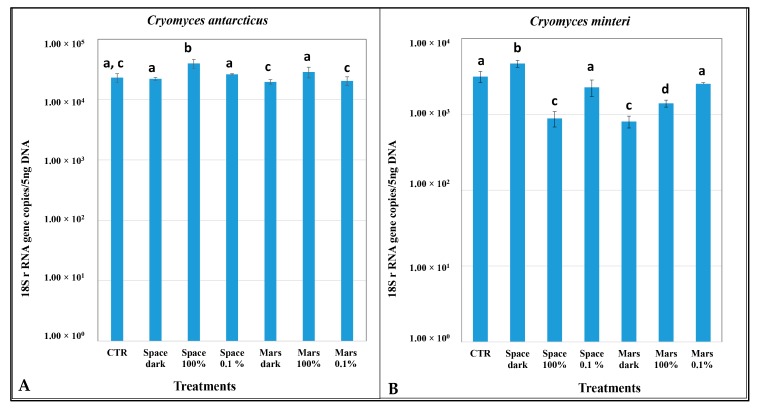
Number of the target gene copies according to real-time q-PCR of different treatments (see Table 1) in (**A**) *C. antarcticus* and (**B**) *C. minteri*. For different columns which have a common letter-designation above, this indicates that the values are not statistically significantly different from each other according to the *t* test (*p* ≤ 0.05). CTR = Ground control; Space dark = space-exposed dark treatment; Space 100% = space-exposed 100% irradiation; Space 0.1% = space-exposed 0.1% irradiation; Mars dark = simulated Mars conditions, dark treatment; Mars 100% = simulated Mars conditions 100% irradiation; Mars 0.1% = simulated Mars conditions 0.1% irradiation.

**Figure 9 life-08-00023-f009:**
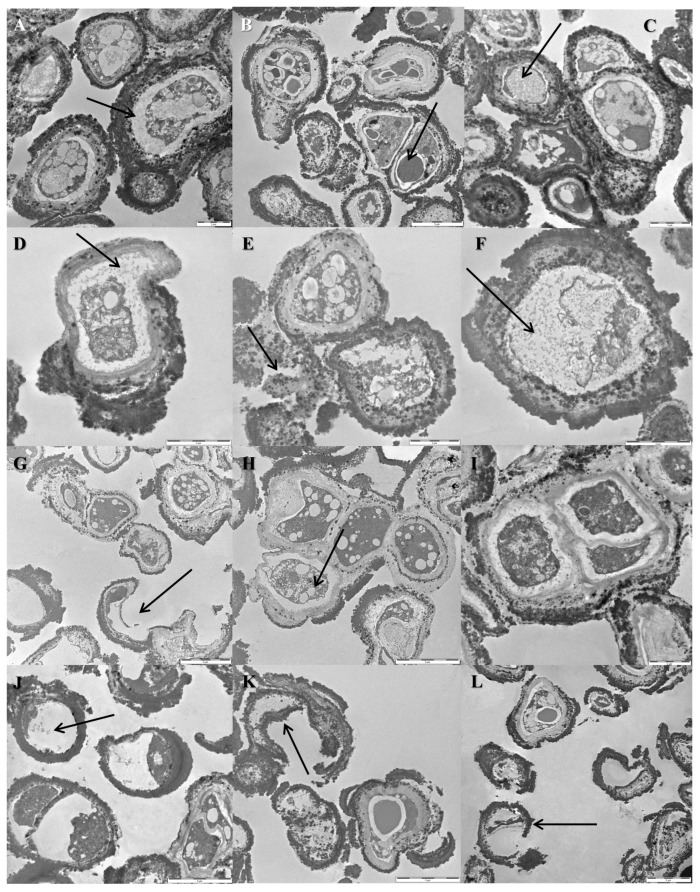
Cells observed by TEM of the *C. antarcticus* Ground control (**A**,**B**), Space 100% irradiation (**C**–**F**) and *C. minteri* Ground control (**G**,**H**), Space 100% irradiation (**I**–**L**). Scale bars = 5 μm for (**B**,**G**,**H**,**J**–**L**) and 2 μm for (**A**,**C**–**F**,**I**).

**Table 1 life-08-00023-t001:** Exposure conditions during the 1.5-year life experiment outside the ISS.

Experimental Treatments	Environmental Parameters			
	Atmosphere within the EXPOSE-E Facility *	Full Solar Radiation; MJ/m² **	Solar Ultra-Violet Radiation; 200–400 nm MJ/m² *	Ionizing Radiation; mGy ***
Space dark	Vacuum (10^−7^ to 10^−4^ Pa)	0	0	238 ± 10
Space Filtered solar ultra-violet radiation (0.1% of full intensity)	Vacuum (10^−7^ to 10^−4^ Pa)	6.49	0.92	238 ± 10
Space solar ultra-violet radiation (100%)	Vacuum (10^−7^ to 10^−4^ Pa)	4369	634	238 ± 10
Simulated Mars dark	95% CO_2_ atmosphere, 1000 Pa	0	0	170 ± 4
Filtered solar ultra-violet radiation (0.1% of full intensity)	95% CO_2_ atmosphere, 1000 Pa	4.18	0.63	170 ± 4
Simulated Mars solar ultra-violet radiation (100%)	95% CO_2_ atmosphere, 1000 Pa	3569	475	170 ± 4

Data were obtained from [12] *, [25] ** and [26] ***. Temperature range during the mission was −21.7–+42.9 °C [12].

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
