# Peer review of "Integrity of the DNA and Cellular Ultrastructure of Cryptoendolithic Fungi in Space or Mars Conditions: A 1.5-Year Study at the International Space Station"

_life, 2018, doi:10.3390/life8020023_

Round 1

Reviewer 1 Report

The paper presents results of experiments exposing the black Micro-Colonial Fungi (MCF) Cryomyces antarcticus and C. minteri to the space environment and to Mars simulated conditions using the EXPOSE-E facility aboard the ISS. Overall this is a good paper and should be published.  There are, however, English syntax errors in the manuscript that need to be corrected.  The experimental protocol could be made to read more clearly if the English were improved (i.e., the wording is a bit awkward). The same is true for the Results section. In the discussion section regarding gene amplification it is stated "these data provided a clear indication of the different resistance of the two fungal strains".  Some suggestion as to why this result is observed is warranted.  Additionally, and most importantly there are no real conclusions drawn from the data.  My reading of the paper indicates that some important conclusions can be made, but the authors fail to make them. 

Author Response

REV 2

The paper presents results of experiments exposing the black Micro-Colonial Fungi (MCF) Cryomyces antarcticus and C. minteri to the space environment and to Mars simulated conditions using the EXPOSE-E facility aboard the ISS. Overall this is a good paper and should be published.  There are, however, English syntax errors in the manuscript that need to be corrected. 

The experimental protocol could be made to read more clearly if the English were improved (i.e., the wording is a bit awkward).

The same is true for the Results section. In the discussion section regarding gene amplification it is stated "these data provided a clear indication of the different resistance of the two fungal strains".  Some suggestion as to why this result is observed is warranted.  Additionally, and most importantly there are no real conclusions drawn from the data.  My reading of the paper indicates that some important conclusions can be made, but the authors fail to make them.  

#answer:  English has been revised by a native speaker scientist. Discussion has been modified and improved.

Reviewer 2 Report

Life (ISSN 2075-1729)

Comments and suggestions to authors:

Onofri et. al. reports data on the resistance of C. antarcticus and C. minteri to extraterrestrial space and Mars-simulated conditions with regards to DNA and cell ultrastructural integrity. The authors found clear differences between treatment conditions and species. The results are interesting and valuable for understanding the limits of life. My major criticism is that results could be better discussed and explained in the Discussion.

Line 30: If both species are melanized, how does the authors conclude that melanin is responsible for the difference between species?

Line 59: The authors should consider elaborating in the introduction of both species (ie. general biology, replication rate, life span, type of melanin, genetic relation, phylum, optimal growth conditions, etc…) Perhaps including some images.

Line 203: The authors should consider stating the main findings as the titles for all result sections instead of just using the technique/method used.

Line 211: It might be useful to present the band intensity values as bar graphs to help the reader understand the differences between conditions.

Line 223: Like in panel B and C, authors should also make note about the increase in band intensity in panel E and F for samples Space Dark, Space 0.1%, Mars 0.1% relative to the control group.

Figure 3: Not sure how useful/practical is to present a print screen of windows. Is there another way of presenting these results in a more quantitative, reader-friendly manner?

Line 283: Again, authors should note the relative decrease observed in the control samples.

Line 301: Not sure what does the letter code means. The p value is not a percentage.

Line 319: Please include reference for scale.

Line 227-329: The authors should consider elaborating on these ideas. How much “compromised” was the control sample relative to the treated samples? How is “minimal hydration” detrimental”? Alternative to the idea of damage in the control sample, is it possible that there is some stability associated with the treatment that could explain why there is more amplification relative to the control sample? The decrease in DNA amplification observed in the control relative to the treated samples is one of the most apparent findings of the study and the authors should consider discussing this thoroughly. 

Line 338: Please specify what previous analyses are consistent. If talking about the same kind of analysis and results, then the findings presented here would be a confirmation.  

Line 349: At this point it looks like the discussion is limited to summarizing and repeating the result section. Although it is useful, the authors should consider discussing their findings. Examples: Are the rates of mutations expected/surprising based on what it is already known? What aspects of the treatment conditions could explain the effects observed? Are there any limitations worth revisiting in the future? Implications? 

Line 364: How does this relates to the results observed in 3.1 gene amplification, where the control samples showed less amplification?

Line 389-394: Here is an example of poor grammar.

Line 400: Is the thickness of the melanin coat different between species (or melanin content per cell)? Do they exhibit the same type of melanin?  Is it possible that the difference may be related to differences in metabolic activity (i.e. replication, life span)? The authors should consider elaborating on the differences between species.

Line 19: consider specifying, at least once, that this is extraterrestrial space

Line 30: at this point it would have been useful to note that both species are melanotic.

Line 49: sentence too complex and unclear.  

Line 89: “gave up to 12.5% CFUs” please specify what this means. It could be interpreted as a percentage decrease or percentage survival?

Line 378: remove, “including fungi”

Author Response

REV 1

Comments and suggestions to authors:

Onofri et. al. reports data on the resistance of C. antarcticus and C. minteri to extraterrestrial space and Mars-simulated conditions with regards to DNA and cell ultrastructural integrity. The authors found clear differences between treatment conditions and species. The results are interesting and valuable for understanding the limits of life. My major criticism is that results could be better discussed and explained in the Discussion.

Line 30: If both species are melanized, how does the authors conclude that melanin is responsible for the difference between species?

#answer: This is because of the thicker (added in  the text) cell wall in C. antarcticus as explained in lines 67-68. Besides Fig. 9 has been replaced with a more defined one. Differences in the cell wall are now highlighted also in Fig 1 (new).

Line 59: The authors should consider elaborating in the introduction of both species (ie. general biology, replication rate, life span, type of melanin, genetic relation, phylum, optimal growth conditions, etc…) Perhaps including some images.

#answer: These concepts have been added in the introduction with the sentence “The two black fungi Cryomyces antarcticus and Cryomyces minteri are cryptoendolithic and micro colonial  species, that occur within sandstone outcrops of the McMurdo Dry Valleys of Antarctica (Southern Victoria Land) [14]. This environment is typically dry, experiences sub-zero temperatures, and exposed to ultra-violet radiation, so has been used as a terrestrial analogue for martian environments. These two species of Cryomyces are endemic for the Antarctic, both are psychrophilic, grow very slowly, and belong to the class Dothideomycetes but do not yet have a phylogenetic designations at the levels of Order or Family [14]. They both have an idiosyncratic morphology, which includes melanized cell walls, much more evident in C. antarcticus, with visible incrustations (Figure 1).” Line 61-69.

Line 203: The authors should consider stating the main findings as the titles for all result sections instead of just using the technique/method used.

#answer: According to reviewer suggestions the title has been changed with “DNA integrity, assessed via single-gene amplification”. All the others have been changed following the same principle.

Line 211: It might be useful to present the band intensity values as bar graphs to help the reader understand the differences between conditions.

#answer: We tried to display data as bar graphs, but on our opinion it does not add any information or represent a further help to the reader respect to the gels. Therefore, we have chosen to keep gels and explain in more details the results in the text (line 227-247), without adding extra figures (even as supplementary).

Line 223: Like in panel B and C, authors should also make note about the increase in band intensity in panel E and F for samples Space Dark, Space 0.1%, Mars 0.1% relative to the control group.

#answer: As reported in the Discussions, lines 380-384 and Lines 405-413, the justification for a more extensive DNA damage found in the Ground controls could be addressed to a minimal re-hydration.

Figure 3: Not sure how useful/practical is to present a print screen of windows. Is there another way of presenting these results in a more quantitative, reader-friendly manner?

#answer: Our feeling is that this is the most immediate and easy way for the reader to catch the differences. Besides, to reduce the burden of illustrations we have skipped Fig. 5 and 6 and calculation of the mutational load was reported in the text Line 166-168 and  275-276 and 289-294.

Line 283: Again, authors should note the relative decrease observed in the control samples.

#answer: As reported above, it has been discussed in lines 380-384 and Lines 405-413.

Line 301: Not sure what does the letter code means. The p value is not a percentage.

#answer: The percentage has been deleted. The letter code is specified in the figure  captions: “The same letter above the bar indicates that the values are not statistically significantly different from each other according to the t Test”. We prefer to use letters instead of * to highlight also the differences between the treatment and not only the differences of treatments with the control.

Line 319: Please include reference for scale.

#answer: Reference scale has been added.

Line 227-329: The authors should consider elaborating on these ideas. How much “compromised” was the control sample relative to the treated samples? How is “minimal hydration” detrimental”? Alternative to the idea of damage in the control sample, is it possible that there is some stability associated with the treatment that could explain why there is more amplification relative to the control sample? The decrease in DNA amplification observed in the control relative to the treated samples is one of the most apparent findings of the study and the authors should consider discussing this thoroughly.

#answer: It is difficult to discuss this apparent incongruence at the moment. The same was found by different authors with different microorganisms. The most reliable explanation we have at the moment is the slight re-hydration since it is well known that complete de-hydration (as achieved in space) enhances resistance. We have summarized the concept with the sentence “Data in the current study indicated an unexpected high level of damage in Ground controls of both species, sometimes even higher than in treated samples. Besides, this situation is rather frequent and has been encountered in other space (or simulated space) experiments carried out using a range of microbial systems (Billi and Rettberg, pers. comm.). One inescapable hypothesis arising from this finding is that de-hydrated state in space can act as a force for the preservation of microbial cells. We also speculate that, in the partially dehydrated ground-control samples, a minimal level of chemical/metabolic activity takes place and, in the presence of atmospheric oxygen plus traces of water, oxygen reactive molecules (ROS) may be produced. At this low metabolic rate, cells are presumably inefficient in their response to oxidative stress [43]. in the Discussion, Lines 405-413.

Line 338: Please specify what previous analyses are consistent. If talking about the same kind of analysis and results, then the findings presented here would be a confirmation. 

#answer:  The previous published results are on survival (CFU’s number) and the cell-membranes integrity. According to revs’ suggestions, the discussion has been completely re-organized.

Line 349: At this point it looks like the discussion is limited to summarizing and repeating the result section. Although it is useful, the authors should consider discussing their findings. Examples: Are the rates of mutations expected/surprising based on what it is already known? What aspects of the treatment conditions could explain the effects observed? Are there any limitations worth revisiting in the future? Implications?

#answer:  Discussion has been re-written.

Line 364: How does this relates to the results observed in 3.1 gene amplification, where the control samples showed less amplification?

#answer:  As reported above this is difficult to explain but is a recurrent situation in Space experiment, even for different microorganisms. It could be speculated that, in a not absolute de-hydration, as for control samples, a minimal chemical/metabolic activity is running and in the presence of atmospheric oxygen plus traces of water oxygen reactive molecules (ROS) may be produced. The low metabolic rate could be not as efficient for detoxification. This hypothesis, already discussed in Billi et al., 2011, has been discussed in the text (line 405-413).

Line 389-394: Here is an example of poor grammar.

#answer:  Discussion has been re-written and English improved by a native English scientist.

Line 400: Is the thickness of the melanin coat different between species (or melanin content per cell)? Do they exhibit the same type of melanin?  Is it possible that the difference may be related to differences in metabolic activity (i.e. replication, life span)? The authors should consider elaborating on the differences between species.

#answer:  According to the rev. suggestion this issue has been more deeply discussed in lines 385-399.

Line 19: consider specifying, at least once, that this is extraterrestrial space

#answer:  It was already specified in the following sentence that the experiment was performed in Earth Orbit: “…and studied in Low Earth Orbit (LEO), using the EXPOSE-E facility on the International Space Station (ISS)”.

Line 30: at this point it would have been useful to note that both species are melanotic.

#answer:  Abstract has been re-written in a more correct English shape. The melanotic character has been added in line 22.

Line 49: sentence too complex and unclear. 

#answer:  Sentence has been modified as follow: “Simulating space conditions in ground experiments is very useful for testing the resistance of extremophilic microorganisms and biomolecules to specific space parameters, i.e. radiation and vacuum, and combinations of extremes. The complexity of the space environment (microgravity, vacuum, and full solar radiation), however, cannot be reliably reproduced in the laboratory [7]. “ Line 48-52

Line 89: “gave up to 12.5% CFUs” please specify what this means. It could be interpreted as a percentage decrease or percentage survival?

#answer:  Sentence has been modified as follow:  “C. antarcticus, retrieved after a 1.5-year exposure, showed a reasonably good survival rate; with 12.5% of cells viable according to cultivation tests (CFUs), up to 80% of cells with intact membranes (by PMA assay) when exposed to space conditions. Samples exposed to Mars conditions simulated in space indicated 8.4% of cells were viable (from CFUs) and 65.0% with intact membranes. C. minteri was much more affected by the exposure both in terms of membrane damage and survival [18-19].” Line 97-102

Line 378: remove, “including fungi”

#answer:  It has been deleted.

Reviewer 3 Report

The paper is devoted to very interesting and important topic concerning the survival ability of black microcolonial fungi under space and other planet extreme environmental conditions. The methodology used is relevant for the study purposes. But some aspects of the paper need to be improved and clarified.

Title – why only DNA integrity is mentioned if the study examined both DNA and ultrastructural damage?

Material and Methods:

-         description of the conditions to which the tested fungal species were exposed apparently repeats the content of Table 1.

-         I am sure that the paper will be of interest to a wide range of reader. Because of that, it would be useful to explain some aspects. For example, why namely the dehydrated colonies of the tested species were used in the experiment. It would be also informative before the detailed description of the molecular approaches used to give a brief introduction for the purpose of each approach, like “The following methods were employed to reveal the level of DNA damage in the tested fungal species: amplification of the rRNA genes and the RAPD assay (molecular fingerprinting) – to detect damage in the individual gene portions and genome, respectively; sequencing with the subsequent alignment – to detect the intensity of mutations; quantitative PCR – to …”.

-         Figure 5 and 6 need some explanations in the captions: what different colors and their absence mean in the terms of presence-absence of mutations (insertion, deletion).

-         Figure 9: it is unclear why ground control in part A has two letters (a, c) above the column; do horizontal bars on the columns represent standard deviations? For how many repetitions? I think the statistical test used (t-test) should be mentioned in Material and Methods, with the statistical software used.

Discussion

Transmission Electron Microscopy

Apparently, the revealed ultrastructural damage needs more than one sentence discussion.

Line 395: It is unclear why C. minteri is mentioned as a species with the extraordinary resistance, if all analyses showed its rather low resistance to most treatments.  

Author Response

REV 3

The paper is devoted to very interesting and important topic concerning the survival ability of black microcolonial fungi under space and other planet extreme environmental conditions. The methodology used is relevant for the study purposes. But some aspects of the paper need to be improved and clarified.

Title – why only DNA integrity is mentioned if the study examined both DNA and ultrastructural damage?

#answer:  The title has been re-worded.

Material and Methods:

-         description of the conditions to which the tested fungal species were exposed apparently repeats the content of Table 1.

#answer:   Conditions to which fungi were exposed are now reported in Tab. 1 only.

-         I am sure that the paper will be of interest to a wide range of reader. Because of that, it would be useful to explain some aspects. For example, why namely the dehydrated colonies of the tested species were used in the experiment.

#answer:  It is now more clearly explained in lines 135-139.

It would be also informative before the detailed description of the molecular approaches used to give a brief introduction for the purpose of each approach, like “The following methods were employed to reveal the level of DNA damage in the tested fungal species: amplification of the rRNA genes and the RAPD assay (molecular fingerprinting) – to detect damage in the individual gene portions and genome, respectively; sequencing with the subsequent alignment – to detect the intensity of mutations; quantitative PCR – to …”.

#answer:  a brief introduction for the purpose of each approach has been added in different sections of the results (i.e. lines 226)

-         Figure 5 and 6 need some explanations in the captions: what different colors and their absence mean in the terms of presence-absence of mutations (insertion, deletion).

#answer:  According to rev. 1 suggestions these figs have been skipped

-         Figure 9: it is unclear why ground control in part A has two letters (a, c) above the column; do horizontal bars on the columns represent standard deviations? For how many repetitions? I think the statistical test used (t-test) should be mentioned in Material and Methods, with the statistical software used.

#answer:  The letter code is specified in the figure  captions: “The same letter above the bar indicates that the values are not statistically significantly different from each other according to the t Test”. We prefer to use letters instead of * to highlight also the differences between the treatment and not only the differences of treatments with the control.

The test used has been added Lines 203-206.

“Means and standard deviations were calculated. Statistical analyses were performed by one-way analysis of variance (Anova) and pair wise multiple comparison procedure (t test), carried out using the statistical software SigmaStat 2.0 (Jandel, USA).”

Discussion

Transmission Electron Microscopy

Apparently, the revealed ultrastructural damage needs more than one sentence discussion.

Line 395: It is unclear why C. minteri is mentioned as a species with the extraordinary resistance, if all analyses showed its rather low resistance to most treatments.  

#answer:  Discussion in its overall has been re-written